# Clinical and Experimental Determination of Protection Afforded by BCG Vaccination against Infection with Non-Tuberculous Mycobacteria: A Role in Cystic Fibrosis?

**DOI:** 10.3390/vaccines11081313

**Published:** 2023-08-01

**Authors:** Sherridan Warner, Anneliese Blaxland, Claudio Counoupas, Janine Verstraete, Marco Zampoli, Ben J. Marais, Dominic A. Fitzgerald, Paul D. Robinson, James A. Triccas

**Affiliations:** 1Sydney Infectious Diseases Institute, Faculty of Medicine and Health, The University of Sydney, Camperdown, NSW 2050, Australia; swar8492@uni.sydney.edu.au (S.W.); c.counoupas@centenary.org.au (C.C.); ben.marais@health.nsw.gov.au (B.J.M.); 2School of Medical Sciences and Charles Perkins Centre, The University of Sydney, Camperdown, NSW 2050, Australia; 3Department of Respiratory Medicine, The Children’s Hospital at Westmead, Westmead, NSW 2145, Australia; anneliese.blaxland@health.nsw.gov.au (A.B.); dominic.fitzgerald@health.nsw.gov.au (D.A.F.); 4Tuberculosis Research Program, Centenary Institute, Camperdown, NSW 2050, Australia; 5Department of Paediatrics and Child Health, Faculty of Health Science, University of Cape Town, Cape Town 7700, South Africa; janine.verstraete@uct.ac.za (J.V.); m.zampoli@uct.ac.za (M.Z.); 6Red Cross War Memorial Children’s Hospital, South Africa, Rondebosch, Cape Town 7700, South Africa; 7Department of Infectious Diseases, The Children’s Hospital at Westmead, Westmead, NSW 2145, Australia; 8Discipline of Paediatrics and Child Health, University of Sydney, Camperdown, NSW 2050, Australia; 9Children’s Health and Environment Program, Child Health Research Centre, University of Queensland, St Lucia, QLD 4072, Australia

**Keywords:** non-tuberculous mycobacteria, *Mycobacterium abscessus*, cystic fibrosis, BCG vaccination, immune response

## Abstract

*Mycobacterium abscessus* is a nontuberculous mycobacterium (NTM) of particular concern in individuals with obstructive lung diseases such as cystic fibrosis (CF). Treatment requires multiple drugs and is characterised by high rates of relapse; thus, new strategies to limit infection are urgently required. This study sought to determine how Bacille Calmette-Guérin (BCG) vaccination may impact NTM infection, using a murine model of *Mycobacterium abscessus* infection and observational data from a non-BCG vaccinated CF cohort in Sydney, Australia and a BCG-vaccinated CF cohort in Cape Town, South Africa. In mice, BCG vaccination induced multifunctional antigen-specific CD4^+^ T cells circulating in the blood and was protective against dissemination of bacteria to the spleen. Prior infection with *M. abscessus* afforded the highest level of protection against *M. abscessus* challenge in the lung, and immunity was characterised by a greater frequency of pulmonary cytokine-secreting CD4^+^ T cells compared to BCG vaccination. In the clinical CF cohorts, the overall rates of NTM sampling during a three-year period were equivalent; however, rates of NTM colonisation were significantly lower in the BCG-vaccinated (Cape Town) cohort, which was most apparent for *M. abscessus*. This study provides evidence that routine BCG vaccination may reduce *M. abscessus* colonisation in individuals with CF, which correlates with the ability of BCG to induce multifunctional CD4^+^ T cells recognising *M. abscessus* in a murine model. Further research is needed to determine the optimal strategies for limiting NTM infections in individuals with CF.

## 1. Introduction

*Mycobacterium abscessus* is a respiratory pathogen of increasing concern, particularly in patients with underlying lung disease such as cystic fibrosis (CF) and chronic obstructive pulmonary disease (COPD) [1]. Currently, it is estimated that 20% of CF patients infected with non-tuberculous mycobacteria (NTM) are infected with *M. abscessus*, an incidence that is estimated to increase further over time [1]. In many patients, *M. abscessus* causes progressive lung function decline with reduced quality of life [2] and it can be fatal [3], with a 15-year cumulative mortality rate of 51% [4]. It is a contraindication for lung transplantation and associated with worse post-lung treatment outcomes [5].

The high morbidity and mortality associated with *M. abscessus* infection is in part driven by the limited treatment available and poor response to current treatment options. Due to its extensive drug resistance profile, patients infected with *M. abscessus* receive prolonged treatment with multiple antibiotics [6,7]. These treatment regimens are associated with significant adverse effects [8] and are costly to both the individual and the health care system [9]. Given the difficulty in treating *M. abscessus* infection, re-assessment of strategies to prevent infection are critical. Development of a vaccine to prevent or reduce severe disease caused by *M. abscessus* infection is one such strategy that could significantly improve the quality of life of many CF patients, and lower the associated health care burden and costs.

Bacillus Calmette Guérin (BCG) is currently the only approved vaccine to prevent tuberculosis (TB), which remains an important cause of infectious deaths worldwide [10]: TB killed 1.6 million people in 2021, and this number is expected to rise in 2022 following the impacts of COVID-19 service disruption. As an attenuated strain of *Mycobacterium bovis*, BCG has been safely used in humans since 1921 [11], is relatively cheap to manufacture and is globally available [12]. BCG vaccination induces potent neutrophil, macrophage and dendritic cell responses, which in turn elicit a strong Th1 response, evident by the production of high levels of IFN-γ-secreting CD4^+^ T cells in vaccinated individuals [13]. Whilst highly protective against disseminated disease and TB meningitis, its efficacy against pulmonary TB is more variable [ranging from 0–80% in clinical trials] and wanes over time. This may reflect a number of factors including BCG strain type, host population genetics and the impact of previous host exposure to environmental mycobacteria [14].

BCG is known to be cross-protective against a range of NTM infections: it is currently recommended to prevent transmission of *M. leprae*, the causative agent of leprosy [15]. Epidemiological data suggest some protection against Buruli ulcer caused by *M. ulcerans* [16], while BCG vaccination is also associated with reduced rates of *M. avium* infection in some countries [17]. The possible cross-protective effects of BCG have been explored against *M. abscessus* in an ex vivo model using BCG-vaccinated patient peripheral mononuclear blood cells (PBMCs), which showed that BCG-specific T cells inhibited growth of *M. abscessus*, as well as inducing strong IFNγ^+^CD4^+^ T cell responses when stimulated with *M. abscessus* ex vivo [18]. While this suggests the cross-reactivity of BCG and *M. abscessus*, whether and how this translates to protection against *M. abscessus* infection has not been evaluated in either humans or preclinical models.

This study aimed to assess the protective efficacy of BCG vaccination against pulmonary *M. abscessus* infection by (i) using an animal model of pulmonary *M. abscessus* infection to describe the effect of BCG vaccination on infection characteristics and immunological pathways, in comparison to naturally acquired immunity, and (ii) using observational data regarding NTM and *M. abscessus* infection rates in two CF cohorts with and without routine BCG vaccination.

## 2. Materials and Methods

### 2.1. Bacterial Strains

*M. abscessus* strain MA07 used in this murine study was a clinical isolate kindly provided by Professor Vitali Sintchenko, Centre for Infectious Diseases and Microbiology, Westmead Hospital (Sydney, Australia). *M. abscessus* and *M. bovis* BCG Pasteur strain (ATCC35734) were grown in a rolling incubator at 37 °C in Middlebrook 7H9 media (Becton Dickinson, BD, Franklin Lakes, NJ, USA) supplemented with 0.5% glycerol, 0.02% Tween 20, and 10% albumin-dextrose-catalase (ADC) or on solid Middlebrook 7H10 agar (BD) supplemented with oleic acid-ADC.

### 2.2. Animals

Six- to eight-week-old C57BL/6 mice were purchased from Australian BioResources (ABR, Sydney, Australia) and maintained at the Centenary Institute (Sydney, Australia) under specific pathogen-free conditions. All animal work was performed in agreeance with Sydney Local Health District (SLHD) Animal Ethics and Welfare Committee guidelines, which are set in accordance with the Australian Code for the Care and Use of Animals for Scientific Purposes (2013), as described by the National Health and Medical Research Council. All work performed in this study were approved by SLHD Animal Ethics and Welfare Committee under protocol 2018-018. 

### 2.3. Mouse Immunisation and Infection

Mice were immunised subcutaneously at base of tail with 5 × 10^5^ colony forming units (CFU) BCG. For infection with *M. abscessus*, mice were anaesthetised by intraperitoneal (i.p.) injection of Ketamine/Xylazine (80–100 mg/kg) and then infected by intranasal (i.n.) administration of 10^6^ CFU *M. abscessus* in a total volume of 25 µL. For challenge experiments, mice were rested after vaccination with BCG or primary infection with *M. abscessus* for ten to twelve weeks, and then infected with 10^6^ CFU *M. abscessus* i.n. in a total volume of 25 µL. Seven days later, the lungs and spleens were harvested and homogenised before serial dilution to plate on supplemented Middlebrook 7H10 agar plates. CFU were enumerated three to five days later and expressed as log_10_ CFU.

### 2.4. Cell Isolation and Flow Cytometry

Seven days before the final challenge, mice were bled from the lateral tail vein to isolate peripheral blood mononuclear cells (PBMCs) using Histopaque1083 (Sigma-Aldrich, St. Louis, MO, USA) to stratify blood cells. Lung single cell suspensions were obtained seven days after challenge by digesting lung tissue with Collagenase IV and DNase (Sigma-Aldrich) for 30 min in a 37 °C waterbath. The tissue was then dissociated using a gentleMACS dissociator (Miltenyi Biotec, Bergisch Gladbach, Germany), and erythrocytes removed using ACK lysis buffer. Single cell suspensions were stained with marker-specific fluorescently labelled monoclonal antibodies (mAbs) to ascertain immune cell populations (Appendix A). Cells were then stained with Fixable Blue Dead Cell Stain (Life Technologies) and fixed with BD Cytofix/Cytoperm™ kit in accordance with manufacturing protocols. Intracellular staining was also performed, using fluorochrome-labelled specific mAbs (Appendix A). To assess antigen-specific expression of cytokine by T cells, PBMCs or pulmonary cell suspensions were stimulated overnight with 10^7^ CFU/mL of *M. abscessus*, then incubated with Protein Inhibitor Cocktail (BD) for 5 h. Cells were then stained for surface and intracellular markers as described above. Samples were run on a BD LSR-Fortessa (BD) and analysed using FlowJo™ analysis software (Treestar, Woodburn, OR, USA) using the gating strategy outlined in Appendix A.

### 2.5. Comparison of BCG Vaccinated and Unvaccinated CF Cohorts

Across a three-year period (2017–2019), the medical records of all paediatric patients attending CF clinics at The Children’s Hospital at Westmead (CHW, Sydney, Australia—no at-birth BCG vaccination) and Red Cross War Memorial Children’s Hospital (RCWMCH, Cape Town, South Africa—universal routine at-birth BCG vaccination) were examined. Ethics approval was granted by the ethics committee at each site (2019/ETH13741 or HREC 249/2020). Both CF clinics have NTM specific screening strategies in place to detect and identify NTM infection based on annual surveillance on sputum samples. The following data were collected from the medical record: demographic data including age, anthropometry and BCG vaccination status. Height, weight and BMI were defined as the highest value for each calendar year (as z-scores), and then as the highest “overall” value for the 3-year period, using WHO (for children aged <2 years) or CDC (for those ≥2 years). Annual lung function (FEV_1_) was defined as the highest value in the calendar year, as %predicted using GLI reference equations. Microbiology status was categorised in each year for NTM, *Pseudomonas aeruginosa*, *Staphylococcus aureus*, *Aspergillus* spp. and *Mycobacterium tuberculosis*: as either “intermittent”, defined as <3 laboratory isolation during the period, “colonised”, defined as ≥3 isolates/year or >50% of samples if >4 samples/year, or “any isolation” which was a positive result in either category. For the entire 3-year period, any positive category in a given year translated to a positive value for “overall” status.

### 2.6. Statistical Analysis

Statistical analysis was obtained using GraphPad Prism 9 software (GraphPad Software, San Diego, CA, USA). Parametric data were described as mean (SD). Pearson’s chi^2^ test was used to compare rates for categorical data and independent *t*-tests for continuous data. Differences between more than two groups were evaluated using one-way analysis of variance (ANOVA). Statistical significance was defined as *p* ≤ 0.05. 

## 3. Results

### 3.1. Protective Immunity Afforded by BCG Vaccination in Mice

To determine if BCG could impact on *M. abscessus* infection, mice initially were either vaccinated subcutaneously (s.c.) with BCG or infected via the intranasal (i.n.) route with *M. abscessus* (as a control for natural immunity acquired from prior exposure, hereafter referred to as convalescent) and the frequency of *M. abscessus*-specific T cells examined. BCG-vaccinated mice displayed significantly higher proportions of multifunctional CD4^+^ T cells secreting IFN-γ, IL-2 and/or TNF compared to naïve mice (Figure 1A,B). Convalescent mice also had notably increased proportions of IFN-γ^+^IL-2^+^TNF^+^CD4^+^ T cells in the blood, though this did not reach significance (Figure 1B). Thus BCG vaccination results in significant levels of circulating *M. abscessus*-specific polyfunctional CD4^+^ T cells in mice.

The impact of BCG-induced immunity on protective efficacy was examined by challenging vaccinated mice i.n. with *M. abscessus*. Prior infection with *M. abscessus* (convalescent group) resulted in significant reduction in bacterial load against subsequent *M. abscessus* infection in both the lung (Figure 2A) and spleen (Figure 2B) compared to unvaccinated mice (~1 log_10_ CFU protection in lung, ~2 log_10_ CFU in spleen), indicating that protective immunity to *M. abscessus* can be achieved in this model. While there was some protection afforded by BCG vaccination in the lung (0.33 log_10_ CFU), this did not reach statistical significance. However, BCG vaccination resulted in significant protection against *M. abscessus* in the spleen, with a reduction of ~1.5 log_10_ CFU compared to unvaccinated mice (Figure 2B). Thus, BCG could impart some level of protective immunity against *M. abscessus* in a murine model, which was most apparent in limiting dissemination of infection.

### 3.2. Adaptive Immune Cell Response after BCG Vaccination and Protection against M. abscessus Challenge in Mice

To determine if the recruitment, expansion, and activation of adaptive immune cells subsets correlated with the protection observed, the cellular response was examined in vaccinated and *M. abscessus* challenged mice. Analysis of CD4^+^ T cells in the lung revealed that overall numbers following *M. abscessus* challenge did not differ between BCG-vaccinated mice compared to convalescent mice (Figure 3A–C). However, there were significantly increased lung CD4^+^ T cell numbers in re-infected mice compared to BCG-vaccinated or unvaccinated animals (Figure 3B). There were no significant differences in CD8^+^ T cell number in either BCG-vaccinated or re-infected mice compared to naïve controls. Interestingly, there were significantly higher levels of B cells in the lungs of BCG-vaccinated mice, which was not seen in mice previously infected with *M. abscessus* (Figure 3B).

When the phenotype of CD4^+^ T cells in the lung following *M. abscessus* challenge was assessed, both vaccinated groups showed significantly increased T-bet expression (Figure 4A,B). There was no significant difference in Rorγt expression between groups (Figure 4A,C). When Th1 effector cytokine production was examined, there was increased IFN-γ, IL-2 and TNF production in CD4+ T cells of re-infected mice (Figure 4D–H). In contrast, CD4^+^ T cells from BCG-vaccinated mice elicited intermediate level of IFN-γ, IL-2 or TNF, which did not differ significantly to levels seen in unvaccinated mice. Thus, BCG vaccination in this model induced Th1-like response within the lung of immunised animals; however, responses were noticeably smaller in magnitude compared with mice previously exposed to *M. abscessus*. Convalescent mice also displayed significantly increased proportions of IFN-γ^+^TNF^+^CD4^+^ T cells in the lungs.

The presence of memory-like CD4^+^ T cells in the lungs of vaccinated mice was also assessed. Effector memory cells, characterised as CD44^hi^CD62L^lo^ circulate in the peripheral tissues, whereas central memory cells (CD44^hi^CD62L^hi^) home to lymphoid organs [19]. In the lungs following *M. abscessus* challenge, both BCG-vaccinated and previously infected groups had significantly higher levels of effector memory-like CD4^+^ T cells (Figure 5A,B). Central memory-like populations were not detected in appreciable numbers in any if the groups at the time point examined. Thus, these data demonstrate that BCG-vaccination and previous *M. abscessus* exposure induces significant effector memory-like populations in the lung.

### 3.3. Comparison of NTM Infection in Relation to BCG Vaccination Status

The Cape Town CF cohort consisted of 91 and the Sydney cohort of 231 children. Characteristics of those children attending both clinics are summarised in Table 1. Comparing the cohorts, Sydney children were taller (0.06 vs. −0.57 peak height z score, *p* < 0.001) than those in Cape Town, but had similar with BMIs (0.27 vs. 0.04 peak BMI z score, *p* = 0.67) lung function (98.7 vs. 91.3 peak FEV1% predicted, *p* = 0.31). 

Table 2 details pathogen categorisation between the cohorts. The overall rates of NTM sampling during the three-year period were equivalent between cohorts: 85% of patients for Cape Town vs. 84% of patients for Sydney (*p* = 0.87). “Overall” values for the entire 3-year period for each of the NTM categories were all numerically lower in the Cape Town cohort and reached statistical significance for rates of NTM colonisation (0.0 vs. 6.7%, *p* = 0.02) and *M. abscessus* colonisation (0.0 vs. 5.7%, *p* = 0.03). Although there were some differences in *P. aeruginosa* and *S. aureus* isolation between in years 2 and 3, there was no significant difference for overall rates of isolation for the other common CF pathogens.

## 4. Discussion

The major aim of this study was to assess the utility of BCG vaccination in protecting against *M. abscessus* infection, using both a murine model and CF cohorts with and without BCG administration. In mice, BCG vaccination did confer significant protection against dissemination of *M. abscessus* to the spleen; however, unlike prior *M. abscessus* infection, minimal protection was afforded by BCG in the mouse lung (Figure 2). BCG is known to be protective against disseminated TB [20]. As BCG is a live-attenuated vaccine delivered systemically, it can seed to the spleen and induce localised immunity, preventing significant dissemination upon subsequent intranasal mycobacterial challenge [21]. Indeed, BCG induced the greatest level of circulating, cytokine-expressing *M. abscessus*-specific CD4^+^ T cells in the blood (Figure 1), yet was a relatively poor inducer of T cell responses in the lung (Figure 3). While both BCG and prior *M. abscessus* infection elicited Th1 responses and multifunctional CD4^+^ T cells in the lung, convalescent mice had higher responses which correlated with the protective efficacy observed. The role of multifunctional CD4^+^ T cells in *M. abscessus* infection has been studied recently in a large cohort study, where triple-positive CD4^+^ T cells correlated with better disease control [22]. The functional capacity of triple-positive CD4^+^ T cells has also been studied in other mycobacterial models, though the evidence is less well-defined. In *M. tuberculosis* infection, polyfunctional CD4^+^ T cells appear to have higher proliferative capacity and are correlated with better disease control [23]. However, in a Phase IIb randomised control trial of the TB vaccine candidate MVA85A, polyfunctional T cells did not correlate with protection against *M. tuberculosis* [24].

Both BCG vaccination and prior *M. abscessus* exposure resulted in significant levels of effector memory-like CD4^+^ T cells in the lung following intranasal challenge. The presence of effector memory-like cells is crucial in the context of vaccine development as these cells are able to rapidly acquire effector functions, subsequently enabling early killing of bacteria and inflammatory cytokine production to recruit other effector cells for early bacterial control [25]. The extensive influx of these cells into the lung following intranasal *M. abscessus* challenge in previously BCG-vaccinated or *M. abscessus*-exposed mice suggests their importance in the early control of *M. abscessus* infection. BCG vaccination was also associated with elevated levels of B cells in the lung following *M. abscessus* challenge, which was not seen after prior *M. abscessus* vaccination. The role of B cells in the context of BCG vaccination against *M. abscessus* has been explored previously. Following BCG vaccination, B cell-deficient mice have overwhelming neutrophilia, which hinders the capacity for DC migration to the mLN in order to prime CD4^+^ T cells to induce a potent Th1 response [26]. Several studies have demonstrated a significant level of antigen-specific IgM and IgG production in the context of BCG vaccination [27] and BCG-specific antibodies have been shown to enhance Th1 responses in the context of *M. tuberculosis* infection [28,29]. Thus, while there is some evidence suggesting a role for B cells in enhancing anti-mycobacterial protective immunity, the data presented here suggest that expansion of BCG-specific B cells does not appear to play a major role in protection against pulmonary *M. abscesses* infection in mice.

In paediatric CF patients in South Africa, where BCG at birth is part of the routine immunisation schedule, lower rates of NTM and *M. abscessus* colonisation were found, as well as lower rates of any isolation of *M. abscessus* compared to those in a CF centre in Australia, where patients have not been vaccinated with BCG. Lower rates of infection in countries with routine BCG vaccination is also supported by a recent report of incidence in a Turkish CF centre where only 2.1% had at least one NTM positive culture from respiratory samples collected between 2012 and 2020 [30]. While epidemiological observations such as this correlate with the notion of the cross-protection of BCG and NTM, it is important to consider some confounding factors that may skew the interpretation of these data. The overwhelming prevalence of *M. tuberculosis*, as well as a higher prevalence of ubiquitous (or environmental) mycobacteria in South Africa may contribute to the cross-protective immunity observed [31]. This theory is supported by evidence that prior sensitisation to NTMs can adversely affect protection afforded by BCG against pulmonary TB [32]. In a worldwide study of broader NTM prevalence by the NTM Network European Trials framework (NTM-NET), *M. abscessus* accounted for 0–2% of all NTM isolates in South Africa, compared with greater than 8% of isolates in Australia [33]. While beyond the scope of this study, these confounders could be addressed using a randomised control trial of BCG vaccination to prevent *M. abscessus* infection across multiple centres with differing rates of *M. abscessus* infection.

## 5. Conclusions

In conclusion, BCG vaccination induced multifunctional antigen-specific CD4^+^ T-cells in the lung and circulating in blood, which correlated with protection against bacterial dissemination to the spleen in a *M. abscessus* mouse model. Observational data across two CF cohorts demonstrated lower rates of NTM and *M. abscessus* infection in the cohort that received routine BCG vaccination. This study provides a platform for future evaluation of BCG, or modified forms of BCG, as a valuable tool to provide protection against difficult to treat *M. abscessus* infection in children with CF.

## Figures and Tables

**Figure 1 vaccines-11-01313-f001:**
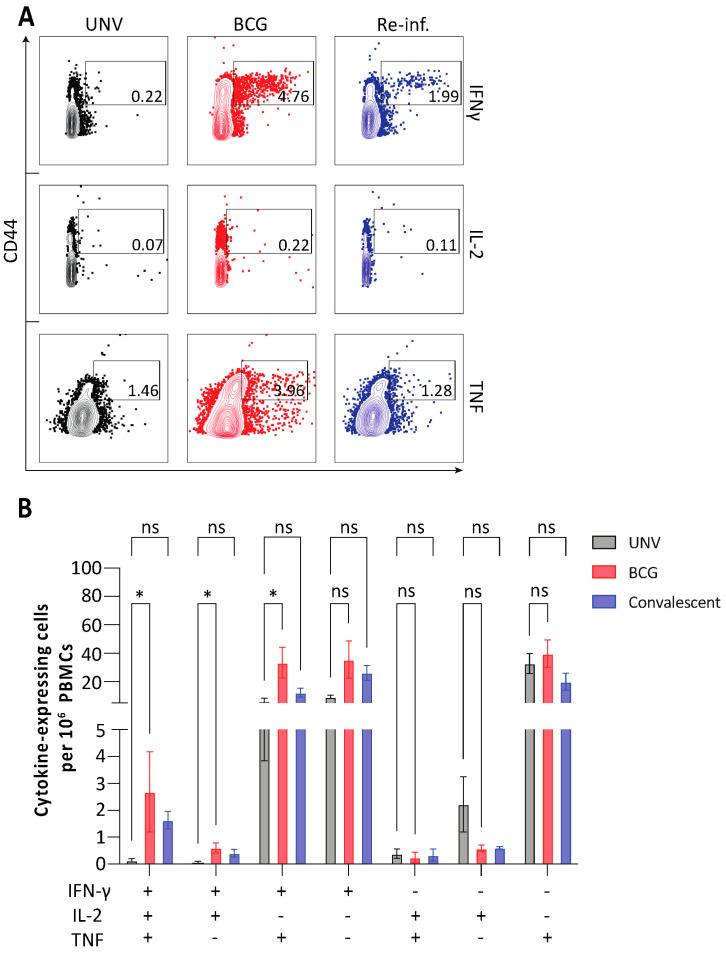
Cytokine production by circulating CD4^+^ T cells following BCG vaccination. C57BL/6 mice (*n* = 6) were vaccinated s.c. with 10^6^ CFU BCG and ten weeks after vaccination, PBMCs were restimulated ex vivo with 5 × 10^5^ CFU *M. abscessus*. IFN-γ, IL-2 or TNF was assessed by flow cytometry. (**A**) Representative FACS plots of cytokine-expressing CD4^+^ T cells in the blood. (**B**) Proportion of CD4^+^ T cells expressing single, double or triple positive combinations of cytokines as determined using Boolean gating. Data are representative of two independent experiments and are shown as mean ± SEM. Differences between BCG-vaccinated and previously infected groups compared to unvaccinated controls was determined by two-way ANOVA (* *p* < 0.05, ns = not significant).

**Figure 2 vaccines-11-01313-f002:**
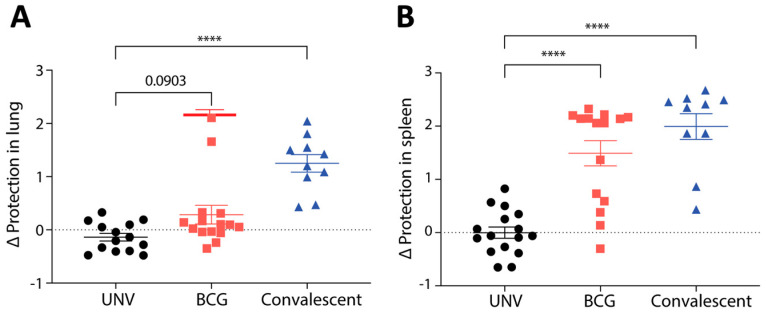
Protection conferred by BCG against *M. abscessus*. C57BL/6 (*n* = 6) mice were vaccinated s.c. once with 10^6^ CFU BCG, infected i.n. with 10^6^ CFU *M. abscessus* or left unvaccinated. Twelve weeks after vaccination, mice were challenged intranasally with 10^6^ CFU *M. abscessus*. Seven days post-infection, bacterial load in the (**A**) lungs and (**B**) spleen was enumerated. Data were collected from two independent experiments and are represented as reduction in log_10_ CFU compared to unvaccinated mice. Statistical significance was evaluated by one-way ANOVA (**** *p* < 0.0001).

**Figure 3 vaccines-11-01313-f003:**
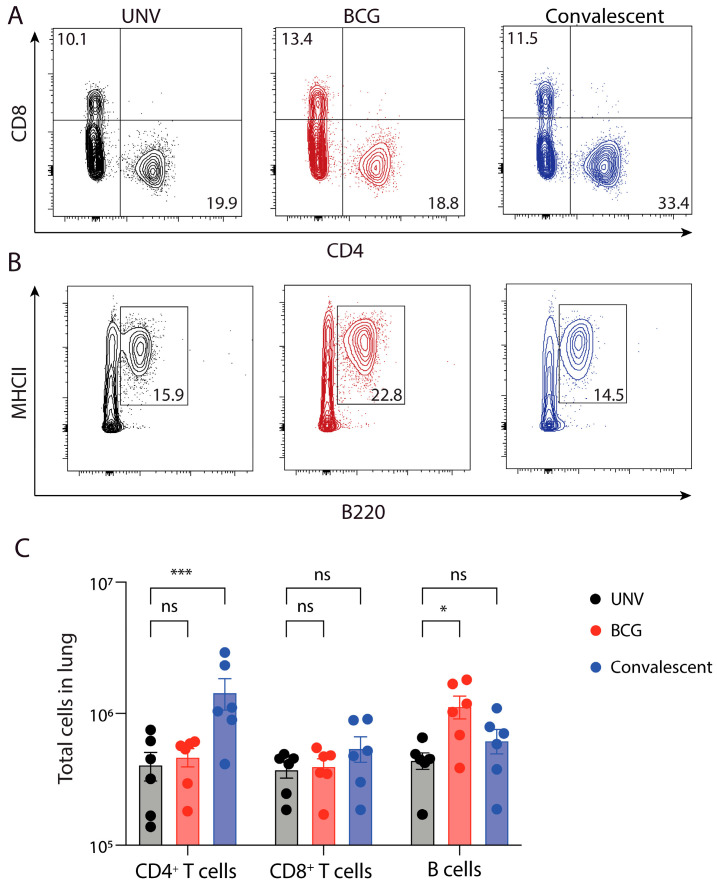
Adaptive immune cell distribution in the lungs of BCG-vaccinated mice challenged with *M. abscessus.* C57BL/6 mice were vaccinated and challenged as in Figure 2 and 7-days post-infection cells were isolated from the lung and analysed using flow cytometry. Representative FACS plots of the proportion of CD4^+^ T cells and CD8^+^ T cells (**A**), or B cells (**B**) in the lung. (**C**) Mean ± SEM of total CD4^+^, CD8^+^ and B cells in the lung following vaccination. Data are representative of two independent experiments (*n* = 5–6) and statistical differences were evaluated using two-way ANOVA (* *p* < 0.05, *** < 0.001, ns = not significant).

**Figure 4 vaccines-11-01313-f004:**
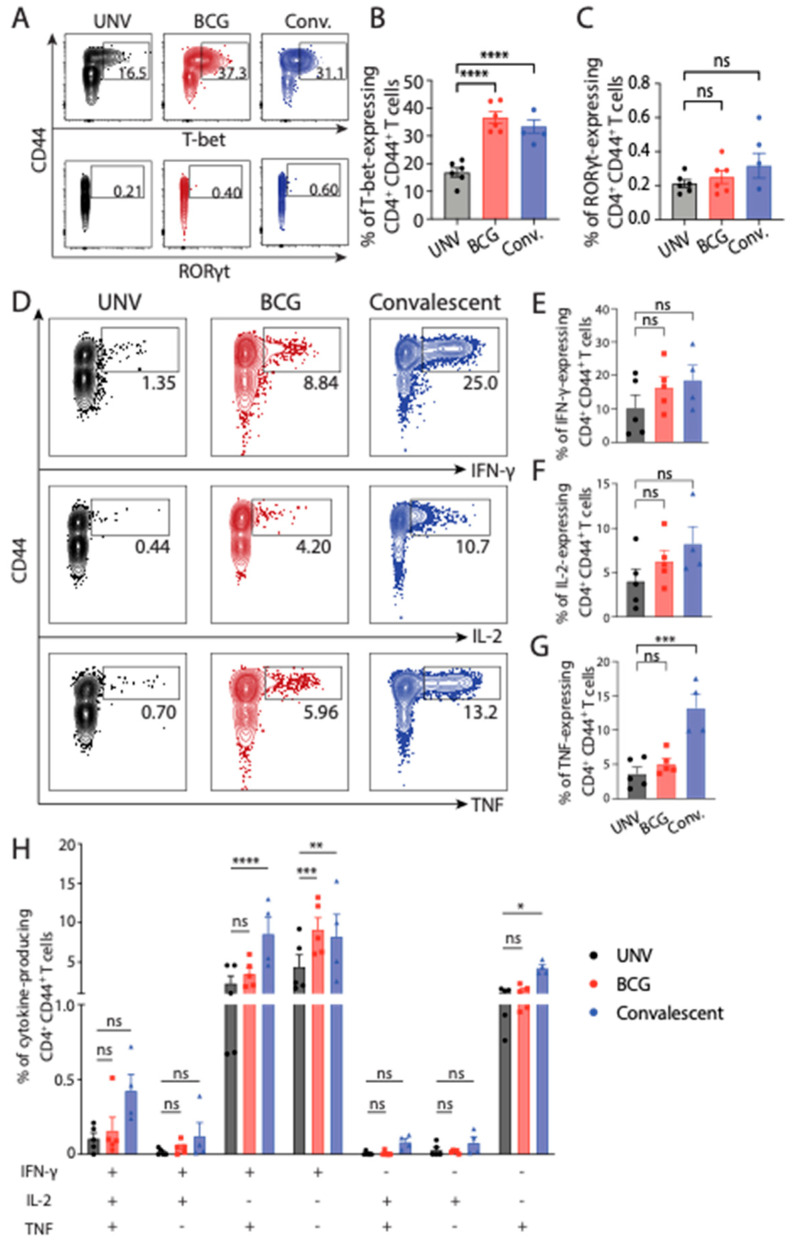
Transcription factor and cytokine production by CD4^+^ T cells in the lungs of BCG-vaccinated mice after *M. abscessus* challenge. C57BL/6 mice were vaccinated and challenged as in Figure 2 and 7-days post-infection cells were isolated from the lung and analysed using flow cytometry. (**A**) Representative FACS plots of the proportion of CD4^+^ T cells expressing T-bet and RORγt. (**B**,**C**) Mean ± SEM CD4^+^ T cells expressing T-bet and RORγt, respectively. (**D**) Representative FACS plots of cytokine-producing CD4^+^ T cells following ex vivo stimulation with 10^5^
*M. abscessus.* (**E**–**G**) Mean ± SEM CD4^+^ T cells producing IFN-γ, IL-2 or TNF, respectively, 7 dpi. (**H**) Mean ± SEM CD4^+^ T cells production multiple cytokines as determined using Boolean gating. Data are representative of two independent experiments (*n* = 5–6) and statistical differences were evaluated using two-way ANOVA (* *p* < 0.05, ** < 0.01, *** < 0.001, **** < 0.0001, ns = not significant).

**Figure 5 vaccines-11-01313-f005:**
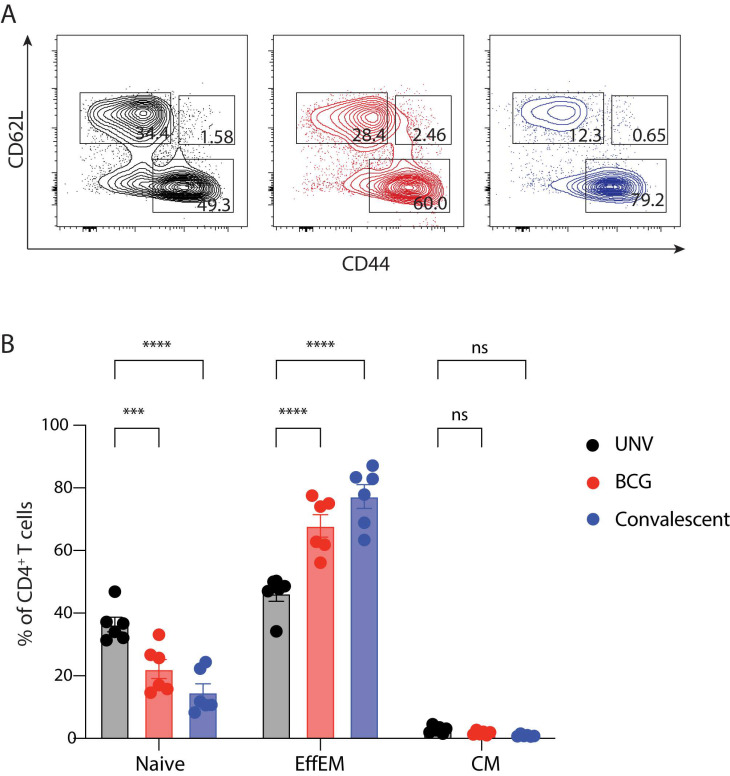
Memory CD4^+^ T cell subsets in the lungs of BCG-vaccinated mice after *M. abscessus* challenge. C57BL/6 mice were vaccinated and challenged as in Figure 2 and 7-days post-infection cells were isolated from the lung and analysed using flow cytometry. (**A**) Representative FACS plots of the proportion of naïve (CD62L^+^CD44^−^), central memory (CM; CD62L^+^CD44^+^), and effector memory (EffEM; CD62L^−^CD44^+^) CD4+ T cells (**B**) Mean ± SEM of memory cell subsets as a percentage of total CD4+ T cells. Data are representative of two independent experiments (*n* = 5–6) and statistical differences were evaluated using two-way ANOVA (*** *p* < 0.001, **** < 0.0001, ns = not significant).

**Table 1 vaccines-11-01313-t001:** Characteristics of Sydney and Cape Town Cystic Fibrosis cohorts.

	RCWMCH ^1^, Cape Town, South Africa				CHW ^2^, Sydney, Australia			
	Year 1	Year 2	Year 3	Overall	Year 1	Year 2	Year 3	Overall
No. patients	71	80	82	91	208	208	213	231
% males	51	49	50	49	51	51	52	52
Age (years)	8.8	8.9	9.7	9.4	9.1	9.4	9.7	9.8
Best ^3^ FEV_1_ (% predicted)	88.0	86.3	85.4	91.3	95.6 *	96.5 *	94.7 *	98.7 *
Best Height (z score)	−0.91	−0.89	−0.86	−0.57	−0.08 *	−0.13 *	−0.09 *	0.06 *
Best Weight (z score)	−0.78	−0.74	−0.72	−0.45	−0.01 *	−0.03 *	−0.03 *	0.18 *
Best ^4^ BMI mean (z score)	−0.22	−0.26	−0.27	0.04	0.06	0.06 *	0.11 *	0.27

^1^ CHW, The Children’s Hospital at Westmead; ^2^ RCWMCH, Red Cross War Memorial Children’s Hospital; ^3^ FEV_1_, Forced expired volume in 1 s; ^4^ BMI, Body Mass Index. * *p* < 0.05. Data collected over the 3-year study period of the study (2017–2019). Data are shown as mean unless otherwise indicated.

**Table 2 vaccines-11-01313-t002:** Comparison of microbiology categorisation of patients between Sydney (Australia) and Cape Town (South Africa) CF cohorts.

		RCWMCH ^1^, Cape Town, South Africa				CHW ^2^, Sydney, Australia			
		Year 1	Year 2	Year 3	Overall	Year 1	Year 2	Year 3	Overall
Number tested for NTM (%)		43 (61)	41 (51)	55 (67)	77 (85)	129 (62)	139 (67)	131 (62)	193 (84)
Any NTM	Intermittent (%)	2.33	4.88	1.82	5.19	4.65	5.04	6.11	7.77
	colonised (%)	0	0	0	0	3.88	5.76	6.11	6.74 *
	any isolation (%)	2.33	4.88	1.82	5.19	8.53	10.79	12.21	11.92
*M. abscessus*	Intermittent (%)	2.33	0	0	1.30	0.78	0	3.05	2.59
	colonised (%)	0	0	0	0	3.10	5.04	6.11	5.70 *
	any isolation (%)	2.33	0	0	1.30	3.88	5.04	9.16 *	7.25
*M. avium*	Intermittent (%)	2.33	2.44	1.82	3.90	3.88	3.60	3.05	4.66
	colonised (%)	0	0	0	0	1.55	1.44	0	1.55
	any isolation (%)	2.33	2.44	1.82	3.90	5.43	5.04	3.05	5.70
*M. intracellulare*	Intermittent (%)	0	4.88	1.82	3.90	0	1.44	0.76	1.55
	colonised (%)	0	0	0	0	0	0	0	0
	any isolation (%)	0	4.88	1.82	3.90	0	1.44	0.76	1.55
Number tested for other bacteria (% sampled)		71 (100)	80 (100)	82 (100)	91 (100)	208 (100)	208 (100)	213 (100)	231 (100)
*Pseudomonas aeruginosa*	Intermittent (%)	28.17	18.75	29.27	45.05	19.71	14.90	15.49 *	35.50
	colonised (%)	16.90	18.75	14.63	28.57	13.94	12.02	9.39	19.05
*Staphylococcus aureus*	Intermittent (%)	64.79	66.25	67.70	84.62	74.04	78.85 *	80.75 *	90.48
*Haemophilus* spp.	Intermittent (%)	12.68	15.00	12.20	27.47	16.83	10.58	11.74	28.14
*Aspergillus* spp.	Intermittent (%)	25.35	21.25	20.73	32.97	22.60	25.48	23.00	37.66

Footnote: ^1^ CHW, The Children’s Hospital at Westmead; ^2^ RCWMCH, Red Cross War Memorial Children’s Hospital; NTM, Nontuberculous mycobacteria. * *p* < 0.05. Data collected over the 3-year study period of the study (2017–2019). Data is shown as mean unless otherwise indicated.

## Data Availability

The data that support the findings of this study are available from the corresponding author upon reasonable request.

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
