# Peer review of "Clinical and Experimental Determination of Protection Afforded by BCG Vaccination against Infection with Non-Tuberculous Mycobacteria: A Role in Cystic Fibrosis?"

_vaccines, 2023, doi:10.3390/vaccines11081313_

Round 1

Reviewer 1 Report

In the manuscript " Clinical and experimental determination of protection afforded by BCG vaccination against infection with non-tuberculous mycobacteria: a role in Cystic Fibrosis?" by Sherridan Warner and colleagues, the authors assessed the protective efficacy of BCG vaccination against pulmonary M. abscessus infection using an animal model of pulmonary M. abscessus infection and observational data regarding NTM and M. abscessus infection rates in two CF cohorts with and without routine BCG vaccination. The authors claim that BCG vaccination induced multifunctional antigen-specific CD4+ T-cells in the lung and correlated with protection against bacterial dissemination to the spleen in the M. abscessus mouse model. Also, the authors claim that observational data across the CF cohorts demonstrated lower rates of NTM and M. abscessus infection in the cohort that received routine BCG vaccination. The manuscript is relevant, well written, the references are up to date. I have no major concerns about this manuscript.

Minor concerns and comments:

In table 2, the number tested for other bacteria, Staphylococcus aureus, show significant differences reduction in the Sydney cohort, compared to the Cape Town.

The paper Wilkie, M. et al (PMID: 35552463), the authors studied the non‑specific effects of BCG vaccination against four bacterial strains, in a randomised controlled clinical study. In the manuscript, the authors describe the unspecific effects of BCG against Staphylococcus aureus. Could the authors please comment on the possible effects described in this paper and how they correlate with the observational study described in this manuscript?

Author Response

Comment: In table 2, the number tested for other bacteria, Staphylococcus aureus, show significant differences reduction in the Sydney cohort, compared to the Cape Town.

The paper Wilkie, M. et al (PMID: 35552463), the authors studied the non-specific effects of BCG vaccination against four bacterial strains, in a randomised controlled clinical study. In the manuscript, the authors describe the unspecific effects of BCG against Staphylococcus aureus. Could the authors please comment on the possible effects described in this paper and how they correlate with the observational study described in this manuscript?
Reply: We thank the reviewer for their comments. While there were some differences in the colonisation rates of Staphylococcus aureus between cohorts, this was only observed at certain timepoints and there was no overall difference (see Table 2). However as suggested by the reviewer, we have updated the results section to better explain our findings (page 8, line 272). In addition to the Wilkie et al article, there are a number of published studies that examine the nonspecific impact of BCG on the control of Staphylococcus aureus infection. However as we observed no overall impact of BCG on Staphylococcus aureus colonisation in our study, we decided to focus our analysis and discussion on M. abscesses.

Reviewer 2 Report

In this manuscript authors assesed the protective efficacy of BCG vaccination against M. abscessus using animal model. They further performed an observational study on cystic fibrosis cohrts with and without BCG vaccination. This is a well design study and neatly written manuscript. I recommend to accept in present form.

Author Response

Comment: In this manuscript authors assesed the protective efficacy of BCG vaccination against M. abscessus using animal model. They further performed an observational study on cystic fibrosis cohrts with and without BCG vaccination. This is a well design study and neatly written manuscript. I recommend to accept in present form.
Reply: We thank for the reviewer for their supportive comments. 

Reviewer 3 Report

"Clinical and experimental determination of protection afforded by BCG vaccination against infection with non-tuberculous mycobacteria: a role in Cystic Fibrosis?" is an article about the capability of BCG vaccination to cross-protect patients with cystic fibrosis from Mab infections. It describes the experiments conducted on mice to verify this hypotesis, and the analysis of two groups of patients in different Austrilian hospitals to further evaluate it. 

The article is well written and easily understandable, and data are convincing. Anyway, I have few concerns about this study that I would like to be elucidated by the authors:
1) the oberserved cross-protection seems to be statistically not relevant in most of the obtained data observing the diagrams, and as you state "In mice, BCG vaccination did confer significant protection against dissemination of M. abscessus to the spleen, however unlike prior M. abscessus in fection, minimal protection was afforded by BCG in the mouse lung (Figure 1)" in line 286-7, and I agree with you. Just very small differences are observable. In this context shouldn't it be better to evaluate the bacterial burden in vaccinated and non-vaccinated cases to have a better idea of this cross-protection effect? But also in this case, wt mice will be able to fight Mab spontaneously...

2) As you state in line 192 "BCG induced the greatest level of circulating, cytokine-expressing M. abscessus specific CD4+ T cells in the blood (Figure 1), yet was a relatively poor inducer of T cell 293 responses in the lung (Figure 3)." , could this effect due to the time of analysis? Maybe you gave not enough or waited too much before doing your analysis to observe any significant effect in lungs? Did you try other times?

3) I read the Turkish paper to understand better the differences in situation across the places taken into account (Australia, Africa, Turkeye). Now, the base for my next question is that NTM are environmental, so there could be a different ralationship between NTM species (and other bacteria) in the environment of those different Countries that is reflected in the incidence of NTM infections (in patients with CF and other clinical problems)? 

Following my suggestion to improve the paper:
- research design: I said that the research design can be improved. This is not related to methods, but in the selection of mice model. For what I saw on a fast research online, the used mice model C57BL/6 consists in common research mice. Anyway, if you want to apply your research to patients with CF, which have a peculiar clinical picture, it would be better to use one or more mice model that could be more similar to the clinical picture of pwCF. Indeed, different more appropriate models are available. Please take this comment into consideration for your next studies, since the other analysis in patients completely bypass the problem of the used model . What I mean is that I don't find the mice data fitting the perfect experimental design to test your hypotesis.

-statistical analysis: please don't use SEM. In statistic is better to use the standard deviation (SD) than SEM as suggested for example here: doi:10.4103/0253-7613.70402. So I must really ask you to improve your data changing SEM with SD for a better statistical analysis and graphical visualization, even if this may give worse graphical visualization.

-bibliography: please uniform your citations since some have the first capital letter in every word while others not (e.g. cit 1 and 3 vs. 2 and 4).

I recommend this article for publication in vaccines after minor revision.

Author Response

Comment 1: the oberserved cross-protection seems to be statistically not relevant in most of the obtained data observing the diagrams, and as you state "In mice, BCG vaccination did confer significant protection against dissemination of M. abscessus to the spleen, however unlike prior M. abscessus infection, minimal protection was afforded by BCG in the mouse lung (Figure 1)" in line 286-7, and I agree with you. Just very small differences are observable. In this context shouldn't it be better to evaluate the bacterial burden in vaccinated and non-vaccinated cases to have a better idea of this cross-protection effect? But also in this case, wt mice will be able to fight Mab spontaneously...
Reply 1: We thank the reviewer for this comment. The data in Figure 2 compares the protection in unvaccinated and vaccinated mice. As indicated in the text, we see strong protection in the mouse spleen in BCG vaccinated mice, but less so in the spleen. This is covered in the Discussion.

Comment 2: As you state in line 192 "BCG induced the greatest level of circulating, cytokineexpressing M. abscessus specific CD4+ T cells in the blood (Figure 1), yet was a relatively poor inducer of T cell 293 responses in the lung (Figure 3)." , could this effect due to the time of analysis? Maybe you gave not enough or waited too much before doing your analysis to observe any significant effect in lungs? Did you try other times?
Reply 2: It is possible that different time points may reveal differences in the immune responses observed, however it is worth noting that data from Figure 1 is pre-vaccination, while Figure 3 data examines responses after vaccaintion then infection. Thus there a number of factors that may influence the differences seen, however we consider a likely explanation is the differing ability of the BCG vaccine to induce immunity at distinct sites of pathogen recognition (see line 292).

Comment 3: I read the Turkish paper to understand better the differences in situation across the places taken into account (Australia, Africa, Turkeye). Now, the base for my next question is that NTM are environmental, so there could be a different ralationship between NTM species (and other bacteria) in the environment of those different Countries that is reflected in the incidence of NTM infections (in patients with CF and other clinical problems)?
Reply 3: We thank the reviewer for their comment, indeed we have addressed this issue in the discussion (line 337) as it is one possible confounder of our study. As indicated in the Discussion, this could be addressed using a randomised control trial of BCG vaccination to prevent M. abscessus infection across multiple centres with differing rates of M. abscessus infection.

Comment 4: research design: I said that the research design can be improved. This is not related to methods, but in the selection of mice model. For what I saw on a fast research online, the used mice model C57BL/6 consists in common research mice. Anyway, if you want to apply your research to patients with CF, which have a peculiar clinical picture, it would be better to use one or more mice model that could be more similar to the clinical picture of pwCF. Indeed, different more appropriate models are available. Please take this comment into consideration for your next studies, since the other analysis in patients completely bypass the problem of the used model. What I mean is that I don't find the mice data fitting the perfect experimental design to test your hypotesis.
Reply 4: We thank the reviewer for this comment, as suggested we will take this into consideration for our next studies.

Comment 5: statistical analysis: please don't use SEM. In statistic is better to use the standard deviation (SD) than SEM as suggested for example here: doi:10.4103/0253-7613.70402. So I must really ask you to improve your data changing SEM with SD for a better statistical analysis and graphical visualization, even if this may give worse graphical visualization.
Reply 5: We commonly use SEM to represent this type of data as it takes into account both the value of the SD and the sample size.

Comment 6: bibliography: please uniform your citations since some have the first capital letter in every word while others not (e.g. cit 1 and 3 vs. 2 and 4).
Reply 6: These errors have been corrected.